# The Combined Cultivation of Feruloyl Esterase-Producing Strains with CMCase and Xylanase-Producing Strains Increases the Release of Ferulic Acid

**DOI:** 10.3390/microorganisms10101889

**Published:** 2022-09-22

**Authors:** Yao Zhang, Zhilin Jiang, Yunran Li, Zhiping Feng, Xian Zhang, Ruiping Zhou, Chao Liu, Lijuan Yang

**Affiliations:** 1College of Bioengineering, Sichuan University of Science & Engineering, Yibin 644000, China; 2Liquor Making Bio-Technology & Application of Key Laboratory of Sichuan Province, Sichuan University of Science & Engineering, Yibin 644000, China; 3Xufu Distillery Co., Ltd., Yibin 644000, China

**Keywords:** ferulic acid, feruloyl esterase, CMCase, xylanase

## Abstract

Feruloyl esterase (FAE)-producing micro-organisms to obtain ferulic acid (FA) from natural substrates have good industrial prospects, and the synergistic effect of multiple bacteria can better improve the yield of FA. In this study, on the premise of the synergistic effect of FAE, hemicellulose, and cellulase, the key strain *Klebsiella oxytoca Z28* with FAE was combined with CMCase and Xylanase-producing strains to produce FA. The combination of strains with higher FA production are *Klebsiella oxytoca Z28*, *Klebsiella pneumoniae JZE*, *Bacillus velezensis G1*, and their FA production can reach 109.67 μg/g, which is 15% higher than that of single bacteria. To explore the effects of temperature, Ph, inoculum amount, distillers grains concentration and fermentation time on the FAE activity of the combination of strains in the fermentation process, and determined that temperature, Ph, and fermentation time were the main influencing factors and optimized through orthogonal design. The optimized fermentation conditions are 34 °C, Ph 8.0, and fermentation days for 6 days, the FAE activity can reach 270.78 U/L, and the FA yield of the combined strain is 324.50 μg/g, which is 200% higher than that of single-strain fermentation.

## 1. Introduction

The massive use of fossil resources has caused huge pollution to the global environment, forcing people to consider the use of sustainable raw materials for the production of industrial raw materials and other necessities. Advanced biological and chemical conversion technologies convert and produce energy, materials, and organic chemicals from biomass feedstocks from different sources, in order to reduce environmental pollution caused by industrial production processes and solve the problem of biomass waste accumulation [1]. Waste lignocellulosic raw materials from agriculture and forestry are good raw materials for their low cost and large quantities [2].

Lignocellulose has a compact structure and is composed of cross-linked cellulose, hemicellulose and lignin [3]. It is very difficult to dehydrolyze lignocellulose by a single enzyme, which requires the joint action of cellulase, hemicellulase, and other coenzymes [4]. Ferulic acid (FA) is a phenolic acid existing in plant cell walls. It has physiological functions, such as antioxidant, antithrombotic, hypolipidemic, prevention and treatment of coronary heart disease, antibacterial, and anti-inflammatory, anti-mutation, and anti-cancer. It is widely used in food additives and in the health products and pharmaceutical industries [5,6,7,8,9]. However, FA exists mainly in the form of ester bonds in plants, such as grain bran, which hinders the utilization of FA. At present, physical and chemical methods can be used to extract FA from plants, which is accompanied by many side reactions and easily causes environmental pollution [10].

Feruloyl esterase (FAE) is the key enzyme to degrade the ester bond between polysaccharide and FA in hemicellulose. FAE derived from different fungi and bacteria were purified and applied to de-starch wheat bran, corn stover, corn cob and bagasse to release FA [1,11,12]. Compared with a single enzyme for substrate decomposition, multi-enzyme synergy can improve catalytic efficiency and increase FA production. When FAE and xylanase work together to extract FA from starch wheat bran, the synergistic FA yield of multiple enzymes is 10 times that of a single enzyme [13]. It is speculated that xylanase first cleaves the backbone of xylan to produce ferulylated xylo-oligosaccharides (FXOS), then FAE can more easily combine with FXOS to release FA [14]. Although using the purified enzyme to produce FA is highly efficient and specific, the pre-purification process is cumbersome, which greatly increases the production cost. Through the co-culture of bacteria and bacteria, it directly acts on the degradation of lignocellulose and shortens the purification process of enzymes. At the same time, the cooperation of multiple bacteria can effectively achieve the degradation of substrates and the generation of products [15,16].

The present study based on the synergistic ability of FAE, cellulase and xylanase, the co-cultivation of enzyme-producing bacteria completed the degradation of lignocellulose and released FA. The key strain *Klebsiella oxytoca Z28(Z28)*, which produced FAE, was co-cultured with the xylanase-producing strain and then co-cultured with the cellulase-producing strain. According to the level of three enzyme activities and the amount of FA production during the co-cultivation period, the optimal strain combination was obtained. A single factor experiment was used to find the main factors affecting the enzyme activity of FAE in the co-culture of three bacteria, and then the orthogonal experiment was used to optimize the fermentation conditions of the co-culture of three bacteria to obtain the optimal release of FA, which provided an experimental basis for the extraction of FA and the industrial application of FAE.

## 2. Materials and Methods

### 2.1. Experimental Materials

Distillers grains are the mixed solids that remain after the fermented grains are distilled out of the wine. The main components of distillers grains include: wheat bran and sorghum. The distillers grains are dried and crushed to a size below 60 mesh. Distillers grains were collected from a winery in Yibin, China; peptone, yeast extract, NaCl, (NH_4_)_2_SO_4_, KH_2_PO_4_, MgSO_4_·7H_2_O, CaCl_2_·H_2_O, and FeCl_3_ were purchased from Kelong Chemical Co., Ltd.(Chengdu, China); Ferulic acid and ethyl4-hydroxy-3-methoxycinnamate were purchased from Yuanye Biotechnology Co. Ltd. (Shanghai, China); The strains used in this study were isolated from the intestines and feces of bamboo rat, Yibin, China, and 16S Rdna was used to classify and identify these bacteria: *Z28*, *Klebsiella pneumoniae JZE*
*(JZE)*, *Bacillus mycoides JEF (JZF)*, *Bacillus cereus JZ3*
*(JZ3)*, *Bacillus velezensis G1 (G1**)*, *Siccibacter colletis G2* (*G2*), and *Bacillus subtilis strain G6 (G6)* [17,18,19,20]. Isolated strains were grown in LB medium, kept at 4 °C, and subcultured periodically for experiments.

### 2.2. Culture Conditions

The enrichment medium used for screening is LB medium; Cellulase-producing bacteria screening medium (g/L): CMC-Na 10, disodium hydrogen phosphate 2.5, potassium dihydrogen phosphate 1.5, peptone 2.5, yeast extract 0.5, Agar 20; Screening medium for xylanase producing bacteria (g/L): peptone 5 g, (NH_4_)_2_SO _4_ 2, MgSO_4_ 7H_2_O 0.25 g, KH_2_PO_4_ 0.5, xylan 10, agar 20; Cellulase enzyme production medium (g/L): CMC-Na 10, dipotassium hydrogen phosphate 1, magnesium sulfate heptahydrate 0.5, ammonium sulfate 2, sodium chloride 2.5, yeast extract 2.5, peptone 5; Xylanase enzyme production medium (g/L): xylan 10, KH_2_PO_4_ 1.0, MgSO_4_·7H_2_O 0.5 g, (NH_4_)_2_SO_4_ 2.0 g, peptone 2.5 g; and feruloyl esterase-producing bacteria screening medium (g/L): (NH_4_)_2_SO_4_ 1.3, KH_2_PO_4_ 0.37, MgSO_4_·7H_2_O 0.25, CaCl_2_·H_2_O 0.07, FeCl_3_ 0.02 g yeast extract 5.0, 10% ethyl4-hydroxy-3-methoxycinnamate 5 mL, pH 6.5. The fermentation medium used for degrading lignocellulose to release FA is composed of (g/L): (NH_4_)_2_SO_4_ 1.3, KH_2_PO_4_ 0.37, MgSO_4_·7H_2_O 0.25, CaCl_2_·H_2_O 0.07, FeCl_3_ 0.02 g yeast extract 5.0, distillers grains 10, pH 6.5. All flasks were incubated at 37 °C under continuous rotary shaker at 180 rpm/min.

### 2.3. Screening and Identification of Strains

After 10 g of bamboo rat intestinal contents was enriched and cultured in LB medium, the enriched solution was diluted and spread in the selection medium for screening cellulase-producing bacteria and xylanase-producing bacteria and then cultured at 37 °C. After 48 h, the grown single cells were picked for separation and purification. A total of 1 mL of activated strain seed medium and 1 mL of 1 mg/mL Congo red solution were added to cellulase production medium and xylanase production medium at 37 °C, 180 r/min. After 7 days of culture, we observed the color change of Congo red in the medium every day, and selected the strain whose color faded faster and then measured its cellulase and xylanase activities by 3.5 and Dinitrosalicylic acid colorimetric method (DNS method). A total of 10 g of distillers grains were added to LB medium (50 mL) for enrichment, then the enrichment solution was diluted, spread in the feruloyl esterase-producing bacteria screening medium, and cultivated in a constant temperature incubator at 37 °C for 72 h, observing the size of the hydrolysis circle produced by the strain on ethyl4-hydroxy-3-methoxycinnamate. The strains with a larger circle diameter were selected, isolated, purified and stored at −80 °C. According to the results of the transparent circle test, strains with larger transparent circles were selected for subsequent screening. It was inoculated into a re-screening medium (50 mL) with 2% inoculum, fermented at 180 rpm and 37 °C for 72 h. A total of 10 mL of fermentation broth was centrifuged at 7000 rpm/min (10 min), and the supernatant was taken as crude enzyme solution. Ethyl4-hydroxy-3-methoxycinnamate was used as the substrate to detect the FAE activity of the strain. The strain with a higher enzyme activity was selected for isolation and identification. The whole genome of the strain was extracted using a bacterial genome rapid extraction kit. PCR amplification was carried out with 16SrDNA universal primers, primer sequences: 27F: 5′-AGAGTTTGATCCTGGCTCAG-3′, 1492R: 5′-TACGGYTACCTTGTTACGACTT-3′, and the products were sent to the biological company for DNA sequence determination. The National Center for Biotechnology Information (NCBI) performed BLAST sequence analysis and comparison and constructed a phylogenetic tree with MEGA7.1 (Appendix A).

### 2.4. Combined Fermentation Culture of Micro-organisms

Different strains of FAE, CMCase, and xylanase were screened in the early stage of the experiment for combined cultivation (there is no antagonism between the strains). The feruloyl esterase-producing strain (*Z28*) and the xylanase-producing strain (*JZE*, *JEF*, *JZ3*) were inoculated into the fermentation medium (1:1) after activation in the LB medium co-culture. The inoculum was 2%. The enzyme activities of FAE, xylanase, and CMCase after 7 days of fermentation were detected, and the yield of FA after 7 days of fermentation was also detected at a wavelength of 320 nm. The above-mentioned optimal bacterial species combination and cellulase-producing strains (*GW1*, *GW2*, *GW6*) were inoculated into the fermentation medium according to the ratio of 1:1 for co-culture fermentation, the enzyme activities of fermentation FAE, xylanase, and CMCase were detected, as well as the yield of FA after 7 days of fermentation, and finally a strain combination with a higher yield of FA was obtained.

### 2.5. Single Factor Experiment

The purpose of this section is to explore the factors that affect the activity of FAE during fermentation by microbial combinations. The optimal bacterial strain combinations *Z28*, *JZE*, and *GW1* were subjected to single factor experiments to explore the effects of temperature, pH, inoculum size, fermentation time, and the addition of distillers grains on FA production. The culture temperature was controlled to be 25 °C, 28 °C, 31 °C, 34 °C, 37 °C, and 40 °C; the initial pH of the medium was adjusted to 4.0, 5.0, 6.0, 7.0, and 8.0; the inoculation amount was 5%, 7.5%, 10%, 12.5%, 15%, and 17.5% inoculated into the fermentation medium. The adjustment gradient of the distillers grains concentration was 2.5 g/L, 5 g/L, 7.5 g/L, 10 g/L, 12.5 g/L fermentation culture, and the fermentation time was controlled to be 4, 5, 6, 7, and 8 d. We detected the changes of FAE activity under various factor conditions.

### 2.6. Orthogonal Experiment

According to the single factor experiment results, temperature, pH, and fermentation time were selected as the influencing factors, and the conditions for increasing the release of FA to the highest level were further searched. Three different levels were selected for these three factors. The design factor level table is shown in Table 1.

### 2.7. Enzyme Activity Determination

For enzyme activity analysis, 10 mL of fermentation broth was centrifuged at 7000 rpm/min (10 min), and the supernatant was collected as the crude enzyme. CMCase activity was determined by spectrophotometry with CMC-Na as substrate. Incubating the crude enzyme solution (1 mL) in 4 mL of 50 mM sodium citrate buffer containing 1% CMC Na (pH 5.0) at 50 °C for 30 min, add 1.5 mL DNS to terminate the reaction, and measure the amount of reducing sugar with a spectrophotometer at 540 nm [21]. One unit (U) of CMCase activity was defined as the required enzyme amount for releasing 1 µmol reducing sugar in 1 min under the standard conditions above. Xylanase activity was determined by spectrophotometry with beech xylan as substrate. The crude enzyme solution (1 mL) was added to 2 mL of 50 mM sodium citrate buffer containing 1.0% beech xylan (pH 5.0), incubated at 50 °C for 10 min, and 2 mL DNS was added to terminate the reaction. The production of xylose was measured with a spectrophotometer at 540 nm [22]. One unit (U) of xylanase activity was defined as the required enzyme amount for releasing 1 µmol xylose in 1 min under the standard conditions above. The enzyme activity of FAE was determined by spectrophotometry with ethyl4-hydroxy-3-methoxycinnamate as substrate. FAE enzyme activity was determined by spectrophotometry with ethyl4-hydroxy-3-methoxycinnamateas substrate. The crude enzyme solution (1 mL) was added to 2 mL of 100 mM Tris HCl (pH 9.0) buffer solution containing 1% ethyl ferulate. After incubation at 37 °C for 2 h, the reaction was terminated in ice bath for 10 min. The resulting FA was measured with a spectrophotometer at 320 nm. The above enzyme activity determination methods take the substrate without enzyme as the control.

### 2.8. Experimental Analysis

After 7 days of fermentation, 70% absolute ethanol (50 mL) in the fermentation broth (50 mL) was extracted in the rotary shaker at 37 °C, 180 r/min for 30 min, and then the extract was taken into a centrifuge tube. After centrifugation at 7000 r/min for 10 min, taking out the supernatant and using a 0.22 μM filter membrane filtration for LC-MS analysis of phenolic acid types and FA content. LC-MS conditions: Waters Acquity UPLC Beh C18 1.7 μm 2.1 mm × 50 mm; Eluent: 10% 0.1% acetic acid water, 90% methanol, column temperature 25 °C, flow rate: 0.2 mL/min; Mass Spectrometry conditions: TEM 500 °C, IS-4500v, CUR 25 psi, GS1 50 psi, GS2 50 psi, detection mode MRM, ferulic acid 192.9 > 134, DP-43v, EP-11V, CE-16V, CXP-5V.

### 2.9. Data Processing

The experiments in this study had three replicates. Data processing used software such as Excel 2016 (Microsoft Corporation, Albuquerque, NM, USA) and SPSS Statistics 25.0 (IBM Corporation, Armonk, NY, USA).

## 3. Results

### 3.1. Combination Culture of Feruloyl Esterase-Producing Strains and Xylanase-Producing Strains

When *Z28* was co-cultured with hemicellulase-producing strains *JZE*, *JZF*, and *JZ3*, we measure the enzymatic activity changes of key enzymes, such as FAE, Xylanase, and CMCase (Figure 1A). It was observed that *Z28* was cultured alone and combined with different strains, the enzyme activity of FAE did not change significantly, and the enzyme activity of FAE remained stable in the range of 300.51~314.86 U/L. We thought that the enzyme activity of FAE of *Z28* during fermentation could be improved by adding a new strain, but we actually observed that co-cultivation had no effect on the FAE activity. In all three combinations, enhancement of the enzyme activity of xylanase and CMCase was detected. The enzyme activity of CMCase of *Z28* combined with *JZF* was the highest, reaching 247.71 U/L; the enzyme activity of xylanase of *Z28* combined with *JZF* was the highest, reaching 283.72 U/L. After examining the changes in the enzyme activities of the combined cultures, we checked the release of FA by the combination of strains (Figure 1B). Compared with single bacterial culture, the FA release of the three different combinations increased by nearly three times. The combination with the highest FA release was *Z28* and *JZE*, whose concentration can reach 150.78 μg/L. Combining the changes in enzyme activity after co-cultivation in Figure 1A with the FA production in Figure 1B, we believe that the presence of CMCase and xylanase can significantly increase the amount of the FA release. Comparing the strain combination *Z28 + JZE* and the strain combination *Z28 + JZF*, the release of FA from the combination of *Z28 + JZE* strains was slightly higher than that of the combination of *Z28 + JZF* strains, which might be caused by the higher cellulase activity in the combination of *Z28 + JZE* strains than in the combination of *Z28 + JZF* strains. Comparing the strain combination *Z28 + JZF* and the strain combination *Z28 + JZ3*, in the process of multi-enzyme synergy, the amount of cellulase and hemicellulase added may affect the release of FA.

### 3.2. Combination Culture of Feruloyl Esterase-Producing Strains and Xylanase and Cellulase Producing Strains

We selected the combination *Z28 + JZE* with higher FA production and the cellulase-producing strain for co-cultivation, and measured the enzymatic activity changes of key enzymes, such as FAE, Xylanase and CMCase (Figure 2A). Compared with the single bacteria, the *G6* was added to the combination *Z28 + JZE*, and the FAE activity was reduced to a certain extent, but its xylanase was the highest level. The cellulase activity of combination of the *Z28 + JEZ + G1* strains combination increased, and the cellulase activity of the combination of *Z28 + JZE + G2* strains combination also increased, but its xylanase activity decreased. For the newly obtained three strain combinations, the production of FA after fermentation was analyzed (Figure 2B). The combination of *Z28 + JEZ + G1* strains and the combination of *Z28 + JZE + G6* strains showed good synergy, and the combination of *Z28 + JZE + G1* strains had the highest FA yield of 210.89 μg/L, which was 2.39 times that of a single strain. Then, the combination of *Z28 + JZE + G2* strains showed reduced FA production. Combining the changes in enzyme activity after co-cultivation in Figure 2A with the FA production in Figure 2B, wee believe that the increase of CMCase and Xylanase enzyme activities in the combination of *Z28 + JZE + G1* strains directly affects the increase of FA production, and the xylanase activity of the combination of *Z28 + JZE + G2* strains combination shows a significant downward trend, which is a direct cause of the decrease in FA production. The reason further explained that the presence of cellulase and xylanase was beneficial to the release of FA.

### 3.3. Single Factor Experiment

Using temperature, fermentation days, inoculum volume, pH, and waste particle concentration as experimental factors, the changes of FAE activity of the combination of *Z28 + JZE + G1* strains were examined to infer the fermentation conditions for high FA production. The effect of different culture temperatures on FAE activity during the fermentation process was explored (Figure 3A). It can be observed that at 28–34 °C, the FAE activity in the fermentation broth increases with the increase of temperature, and when the temperature reaches 34 °C, the FAE activity reaches the maximum value. Then, with the increase of temperature, the enzyme activity of FAE decreased continuously, and the high temperature was not conducive to the growth and metabolism of the three strains, thus affecting the production of FAE. When the fermentation time was between 4 and 6 days, the enzyme activity of FAE was low for three consecutive days, but the FAE showed a sharp increase trend on the 7th day, and the FAE activity decreased on the 8th day (Figure 3B). It is because the nutrients present in the fermentation medium are not unique.

In the early stage of fermentation, the micro-organisms mainly use substances such as yeast powder, amino acids and polysaccharides in the medium. When the amount of these substances is reduced, they degrade and use distillers grains to provide energy for their own growth and metabolism. The effect of the initial inoculum amount of the strain on the FAE activity during the fermentation process was explored (Figure 3C). It can be seen that when the inoculum amount is 6%, the enzyme activity of FAE is the highest. When the inoculum amount was 9%, 12%, 15%, and 18%, the relative enzyme activity of FAE was stable between 60% and 80%. It is speculated that over-inoculation will lead to a decrease in FAE activity. The effect of the initial pH of the fermentation broth on the FAE activity during the fermentation of the strain combination was studied (Figure 3D). When the initial pH of the fermentation broth was in the range of 4.0–7.0 and 9.0, the relative enzyme activity of FAE was stable in the range of 50–70%. When the pH was 8.0, the FAE was significantly higher than other pH levels. It can be concluded that the initial pH of fermentation is 8.0, which is favorable for FAE release. The effect of discard concentration on FAE activity during fermentation was analyzed (Figure 3E). It can be found that the FAE activity of the fermentation broth can maintain more than 70% of the enzyme activity detected by different concentrations of discarded grains. By comparison, it can be seen that the concentration of discarded grains exceeding 10 g/L will inhibit the activity of FAE to a certain extent.

### 3.4. Orthogonal Experiment on Production Rule of Feruloyl Esterase by Strain Combination

The purpose of this part of the study is to obtain the optimal conditions for the highest FAE enzymatic activity of the microbial combination. According to the results of the single factor experiment, it is concluded that five single factors have an effect on the FAE activity of the microbial combination. Among them, the inoculum amount and the concentration of distillers grains are too high to inhibit the FAE activity. The conditions (pH, fermentation time, and temperature) that had a significant and stable effect on FAE activity were selected for orthogonal experiments to optimize the fermentation conditions (Table 2). According to the comparison of R_B_ > R_A_ > R_C_ in Table 2, it can be concluded that the influence condition of FAE activity is fermentation time > pH > temperature. According to the comparison of K_1_, K_2_, and K_3_ values calculated by orthogonal experiments, it is concluded that the optimal fermentation enzyme production condition is fermentation time of 7 days, pH of 8.0, and a temperature of 34 °C. Under the optimized culture conditions, the FAE activity of the microbial combination could reach 270.78 U/L. Compared with the highest FAE activity of 215.94 U/L in Table 2, the optimized FAE activity was increased by 20%.

### 3.5. Comparative Analysis of FA Yield before and after Optimization of Optimal Combination Fermentation Conditions

The production of FA was quantitatively detected by LC-MS compared with the production of FA before and after optimization of single bacteria and fermentation conditions (Figure 4). It can be found that through the co-cultivation of FAE-producing strains and CMCase-producing and xylanase-producing strains, the FA production has been improved to a certain extent, from the previous 95 ug/g to 109.67 μg/g. Through the optimization of the fermentation conditions of the orthogonal experiment, the FA yield of the co-culture can reach 324.50 μg/g, which is 200% higher than that of the single bacterial culture.

## 4. Discussion

Due to the generation time of fungi, it is not recommended to cultivate fungi to release FA in a short period of time [23]. The bacteria with a generation time to release FA also has the problem that the enzyme activity of FAE is low. During the co-cultivation process, the FAE activity did not change significantly, and the co-culture environment had no obvious effect on the FAE activity. However, under the optimal combination, the FA release can be increased by 4 times using co-culture. Compared with Duan’s use of FAE and xylanase to synergize the FA release of de-starched wheat bran by 10 times [13], the increase in FA production in this study was lower. However, the study provides a mixed culture method to improve FA production. The enzymatic activities of cellulase and xylanase increased, and the FA production also increased. This result is consistent with the previously reported results that the synergistic effect of xylanase, FAE, and cellulase improves the hydrolysis efficiency of lignocellulose [24]. In Figure 2, when the activities of the three enzymes were found to be relatively similar, only the microbial combination *Z28 + JZE + G1* had a 4-fold increase in FA production. This result indicates that other enzymes may be included in lignocellulose decomposition and directly increase the production of FA, which may be derived from some other enzymes (α-glucuronidase, acetylesterase) secreted by the *G1* to assist in the decomposition [25].

Temperature and pH are important factors affecting the enzymatic activity of FAE. Zhenshang Xu found that FAE derived from Lactobacillus has the best enzymatic activity at pH 7.0~8.0 [26]. Ziyang WU found that the optimal fermentation conditions for FAE from *Acetobacter* were pH 7.0, temperature 30 °C, inoculum size 10%, and fermentation time of 24 h [27]. Similar to the previous study, the maximum temperature of FAE activity in this study was 34 °C and the optimum initial pH was 8. Adjusting of temperature and initial pH can increase FAE activity during fermentation. The difference is that the highest inoculum amount for the study of enzyme activity is 6%, and an inoculum amount that is too high will reduce the enzyme activity of FAE. In addition, the increase of the concentration of distillers grains will increase the viscosity of the culture, affect the growth of the bacteria during the culture process, and reduce the enzyme activity of FAE. The microbial combination in this study reached the highest value of FAE activity on the 6th day, and studies have reported that the optimal enzyme production cycle depends on the properties of substrates, organic matter, and micronutrients [28].

A previous study showed that the FA extracted from de-starch wheat bran by alkali extraction was 4.04 mg/g, and the FA released by enzymatic reaction was 2.8 mg/g [29]. Xu obtained a maximum release of 199 μg of FA from 0.2 g of de-starched wheat bran using purified FAE [30]. Under the optimized mixed culture conditions in this study, the release amount of FA can reach 324.50 ug/g, which is three times that of single bacterial culture. However, the amount of FA released from co-culture is still lower than that of alkaline extraction and recombinase. Therefore, subsequent studies can use transcriptome sequencing technology to research the FA metabolic pathway of microbial assemblages during co-culture, and reveal the release mechanism of FA. At the same time, a higher release of FA can be obtained by changing the different treatment methods of the substrate or by screening and domesticating new microbial consortia.

## 5. Conclusions

At present, most research on FA production use crop bran as the raw material. Liquor discarded grains also contain crop raw materials, and lignocellulose and other resources are not utilized after liquor production. Using discarded grains as the raw material for FA production solves the problem of the accumulation of discarded grains of liquor and provides a new way of high-value utilization of discarded grains of liquor. This study effectively increased the release of FA by means of mixed bacterial culture and provided a new way to increase the release of FA. In the follow-up, the metabolism of micro-organisms during the co-cultivation process can be specifically studied to reveal the release rule of FA.

## Figures and Tables

**Figure 1 microorganisms-10-01889-f001:**
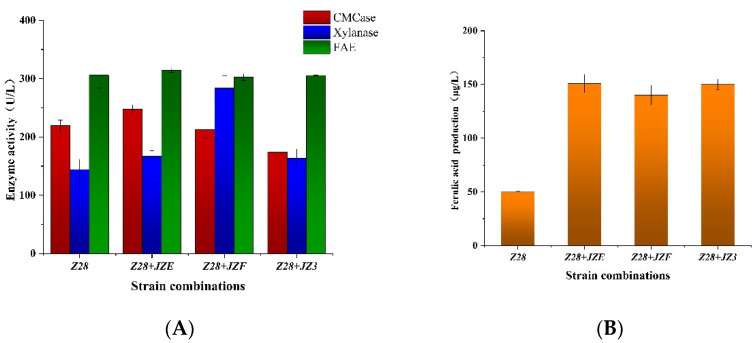
Analysis of enzyme activity (**A**) and FA production (**B**) during the combined culture of feruloyl esterase-producing strains andxylanase-producing strains.

**Figure 2 microorganisms-10-01889-f002:**
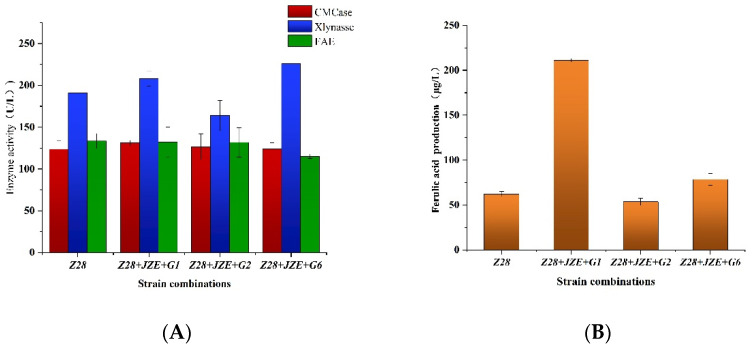
Analysis of enzyme activity (**A**) and FA production (**B**) during the combined culture of feruloyl esterase-producing strains, xylanase-producing strains, and CMCase-producing strains.

**Figure 3 microorganisms-10-01889-f003:**
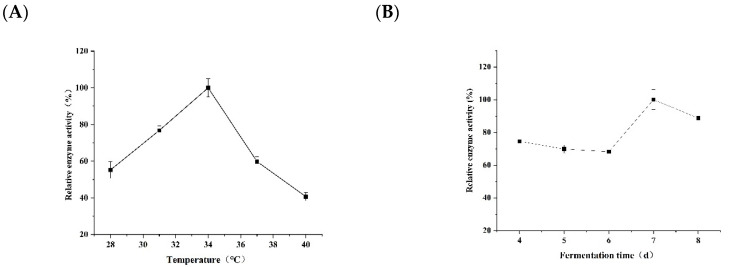
Changes of FAE enzyme activities of microbial combinations under different factors: (**A**) Change laws of FAE enzyme activities of microbial combinations at different temperatures; (**B**) Change laws of FAE enzyme activities of microbial combinations at different fermentation times; (**C**) Different inoculations. The change rule of FAE enzyme activity of the microbial combination under different pH; (**D**) The change rule of FAE enzyme activity of the microbial combination under different distillers grains concentration(**E**).

**Figure 4 microorganisms-10-01889-f004:**
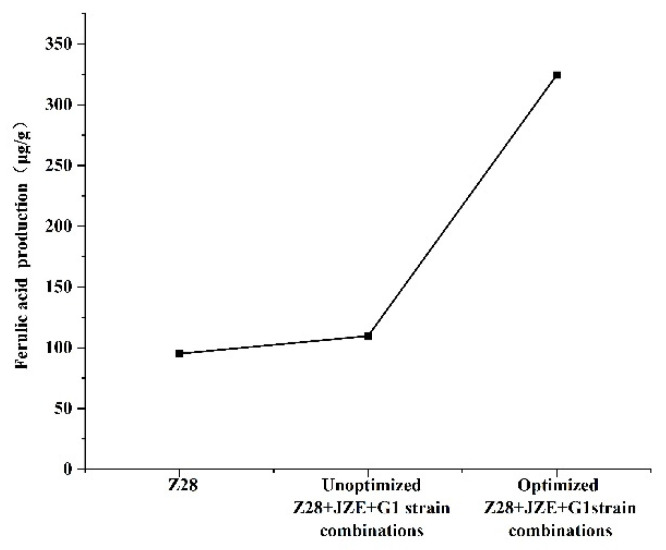
Comparative analysis of FA yield before and after optimization of optimal combination fermentation conditions.

**Table 1 microorganisms-10-01889-t001:** Factors and levels of orthogonal experiments.

Level	Fermentation Time/d	pH	Temperature/°C
1	6	7.0	31
2	7	8.0	34
3	8	9.0	37

**Table 2 microorganisms-10-01889-t002:** The results of the orthogonal experiment.

Test No.	A pH	B Fermentation Time/d	C Temperature/°C	FAE/U/L
1	7	6	31	163.05
2	7	7	34	111.91
3	7	8	37	124.96
4	8	6	34	215.94
5	8	7	37	107.33
6	8	8	31	146.47
7	9	6	37	179.26
8	9	7	31	121.43
9	9	8	34	143.65
K_1_	133.31	186.08	143.65	
K_2_	156.58	113.56	157.17	
K_3_	148.11	138.36	137.18	
R	23.27	72.53	19.98	

## Data Availability

Date is contained within the article.

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
