# Peer review of "The Combined Cultivation of Feruloyl Esterase-Producing Strains with CMCase and Xylanase-Producing Strains Increases the Release of Ferulic Acid"

_microorganisms, 2022, doi:10.3390/microorganisms10101889_

Round 1

Reviewer 1 Report

General comments:

1.      The manuscript is missing lines numbered in the margin.

2.      The manuscript lacks clarity in the sentences. Please rewrite the text.

3.      Instead of "joint culture" for the cultivation of more than one type of microorganism, the authors should use "co-culture" or "mixed culture."

4.      Short names for the strains should be defined for the first time the strain is mentioned in the text in the text, e.g. Klebsiella oxytoca Z28 (Z28).

5.      The name of the microorganism should be italicized (e.g. Klebsiella oxytoca), but not the strain designation (e.g. Z28).

6.      Instead of "biological enzymes" the Reviewer would suggest "enzymes" because the microbial enzymes are biological products of the cell, and it is unnecessary to point this out.

7.      Instead of "shaking table" use "rotary shaker" (Page 4)

Specific comments:

1.      Page 2: "… in order to improve environmental pollution and alleviate the use of non-renewable resources"

Authors should consider rewriting the text. The sentence is hard to understand.

Material and Methods

2.      Page 3,

The substrate for growing the microorganisms is not defined. Distillers grains are a mix of spent grains from a specific type of production. Please describe the substrate (was it dried before it was stored, composition, type of grains, water content, particle size,…)

3.      Page 3 "… ethyl ferulic acid"

Did the authors mean "ethyl ferulate" or "ferulic acid ethyl ester". Please correct the name of the substance accordingly.

4.      Page 3-4:

The authors used several strains of bacteria in the research. Please describe methods used to isolate and purify and genetic methods for identification of the strains (primers, alignment of the specific DNA fragments, …) or provide a reference.

Page 4:

5.      Authors should explain how they select the conditions for the Orthogonal experiment. (add this comment in Results and Discussion)

Results

6.       Page 5:" Comparing the strain combination Z28+JZE and the strain combination Z28+JZF, it is speculated that cellulase is more conducive to improving the release of FA than xylanase."

I do not agree with this statement because the ferulic acid concentration (Fig. 2B) shows that regardless of the enzyme activities, the end concentration of the acid is similar for all three co-cultures (approx. 150 U/L).

Page 6, Figure 2.

7.      Relative similar activities of three enzymes were determined for all four cultures, but the content of ferulic acid for one co-culture was almost 4 times higher. That suggests that some other enzyme might be included in the decomposition of the lignocellulose substrate and directly improve the yield of ferulic acid. For e.g. some enzymes from Bacillus strains participate in lignin degradation.

8.      Page 6: "Then, with the improvement of culture, the activity of FAE decreased continuously, and the high temperature was not conducive to the growth and metabolism of the three strains, thus affecting the production of FAE enzymes."

The sentence is not clear. What did the authors mean by "improvement of the culture"?

9.      Page 7:" It can be seen that the concentration of discarded grains has no significant effect on the FAE activity. By comparison, it can be seen that the concentration of discarded grains exceeding 10g/L will inhibit the activity of FAE to a certain extent."

The Reviewer disagrees with this statement. At 5 g/L, the culture has 100 % activity; at 10 g/L the activity decreases by 20 % . This means that the concentration significantly affects the activity of the FAE. Furthermore, the rheology of the culture could significantly influence culture performance. Instead of homogenous culture (culture grown on LB medium) in this experiment, the substrate is undissolved (solid phase). As the solid fact increases, the viscosity of the culture increases, changing the mixing pattern and mass transfer which influence the growth of microorganisms. Please comment on this in Results and Discussion.

10.   Page 8; "It can be seen that the FAE activity affects the fermentation time > pH > temperature, and the optimal conditions for fermentation enzyme production are 7 days for fermentation time, pH 8.0, and temperature 34 °C."

This conclusion cannot reasonably be drawn from Table 2. The cultivation  under this condition was not conducted (pH=8, T34oC and 7 days)

11.   Page 9-10

The Discussion should not be an extensive literature review (please see the first paragraph of the Discussion). The authors should compare results with the literature data, comment on whether or not this research makes a new contribution to the field,  if it opens up new paths for further research, whether it raises questions that are left unanswered, etc. Please rewrite the Discussion.

12.   Conclusions

Conclusions are lengthy and should be shortened. The authors could also give directions for future research.

Author Response

Thank you for the comments on our manuscript (microorganisms-1914708) titled "Combined Cultivation of Feruloyl Esterase-producing strains with CMCase and Xylanase-producing Strains Increases the Release of Ferulic Acid". We have taken all the comments into consideration, and have revised our manuscript accordingly. We believe that the supplementary contents, revised analyses, and modified presentations have substantially improved the manuscript. For details, please see the attachment.

Reviewer 2 Report

The authors report the increase in FA release 3.02 times, 2.81 times and 3.01 times. However, these results are in the same range of improvement, therefore the authors are suggested to report these results as 3 times increase for all the three conditions. The same comment regards the FA yield from the orthogonal experiment, where the increase reported is 198% higher than that of the single bacterial culture, which would be equal to 200%.

Figure 4 is not informative. The authors are suggested to report the comparison of the improvement in FA yield using their strategies in the text of the manuscript. 

In the discussion section there is no comparison with previous results reported in the same field in the literature. The authors should report how their results impact the research in the same area of work.

Overall, the authors are recommended to improve the presentation of results and conclusions. 

Author Response

Thank you for the comments on our manuscript (microorganisms-1914708) titled "Combined Cultivation of Feruloyl Esterase-producing strains with CMCase and Xylanase-producing Strains Increases the Release of Ferulic Acid". Although the manuscript has shortcomings, the reviewers still gave a positive evaluation. Thank you for your understanding and great suggestions, we have revised the manuscript as per your suggestion For details, please see the attachment.

Reviewer 3 Report

There are several names of microorganisms in the text that are not written in italics.

Author Response

Thank you for the comments on our manuscript (microorganisms-1914708) titled "Combined Cultivation of Feruloyl Esterase-producing strains with CMCase and Xylanase-producing Strains Increases the Release of Ferulic Acid". Although the manuscript has shortcomings in language expression and content analysis, you give high evaluation and understanding, which motivates us to do better. The answers to the specific questions you asked are in the attachment. Thanks again for your help.

Round 2

Reviewer 1 Report

1.      Point 4. Page 3-4:

The authors used several strains of bacteria in the research. Please describe methods used to isolate and purify and genetic methods for identification of the strains (primers, alignment of the specific DNA fragments, …) or provide a reference.

Response 4: The strains producing cellulase and xylanase were isolated and screened from the intestinal tract of bamboo rat. Feruloyl esterase-producing strain is isolated and screened from distillers grains.16S rDNA was used to classify and identify these bacteria. The primer sequence is: 27F:5'-AGAGTTTGATCCTGGCTCAG-3', 1492R: 5'-TACGGYTACCTTGTTACGACTT-3'.The authors provide the references(Reference from 17 to 20) on methods of isolation and purification and genetic methods for identification of the strains.

Reviewer's comment:

The authors should write a new subsection in Materials and Methods describing the procedure for the isolation of strains, purifying the culture, amplification of 16 S rDNA with primers (reference for primes or sequences if authors designed them), PCR reaction condition, purification of DNA and add results of alignment of amplified DNA using alignment search tool such as BLAST. The alignment results should be presented in Results or  Supplementary material if they were not published before. However, if the results were already published, references should be added to the list of microorganisms used in the work.

2.      Discussion

Lines 599-600

"Due to the long growth cycle of fungi, it is not recommended to cultivate fungi to release FA in a short period of time. The bacteria with a short growth cycle to …".

A microorganism's rate of exponential growth is expressed as generation time and doubling time. Therefore, the reviewer suggests using "generation time" or "doubling times" instead of the "growth cycle".

Author Response

Thanks again for your valuable comments, we have revised the manuscript and attached supplementary materials, see the attachment for details.
